# Efficacy of Kan Jang^®^ in Patients with Mild COVID-19: Interim Analysis of a Randomized, Quadruple-Blind, Placebo-Controlled Trial

**DOI:** 10.3390/ph15081013

**Published:** 2022-08-17

**Authors:** Levan Ratiani, Elene Pachkoria, Nato Mamageishvili, Ramaz Shengelia, Areg Hovhannisyan, Alexander Panossian

**Affiliations:** 1The First University Clinic of Tbilisi State Medical University, Gudamakari St., Tbilisi 0141, Georgia; 2Department for History of Medicine and Bioethics, Faculty of Medicine, Tbilisi State Medical University, Vazha-Pshavela Ave. 33, Tbilisi 0162, Georgia; 3Institute of Fine Organic Chemistry, National Academy of Science, Azatutian Ave. 26, Yerevan 375014, Armenia; 4Phytomed AB, Bofinkvagen 1, 31275 Våxtorp, Sweden

**Keywords:** adaptogens, Kan Jang^®^, *Andrographis paniculata*, *Eleutherococcus senticosus*, clinical trial, mild COVID-19, IL-6, inflammatory symptoms

## Abstract

Kan Jang^®^, the fixed combination of *Andrographis paniculata* (Burm. F.) Wall. ex. Nees and *Eleutherococcus senticosus* (Rupr. & Maxim.) Maxim extracts, is a herbal medicinal product for relieving symptoms of upper respiratory tract infections. This study aimed to assess the efficacy of Kan Jang^®^/Nergecov® on duration and the relief of inflammatory symptoms in adults with mild COVID-19. 86 patients with laboratory-confirmed COVID-19 and mild symptoms for one to three days received supportive treatment (paracetamol) and six Kan Jang^®^ (daily dose of andrographolides—90 mg) or placebo capsules a day for 14 consecutive days in this randomized, quadruple-blinded, placebo-controlled, two-parallel-group study. The primary efficacy outcomes were the decrease in the acute-phase duration and the severity of symptoms score (sore throat, runny nose, cough, headache, fatigue, loss of smell, taste, pain in muscles), an increase in cognitive functions, physical performance, quality of life, and decrease in IL-6, c-reactive protein, and D-dimer in blood. Kan Jang^®^/Nergecov^®^ was effective in reducing the risk of progression to severe COVID-19, decreasing the disease progression rate by almost 2.5-fold compared to placebo. Absolute risk reduction by Kan Jang treatment is 14%, the relative risk reduction is 243.9%, and the number Needed to Treat is 7.14. Kan Jang^®^/Nergecov^®^ reduces the duration of disease, virus clearance, and days of hospitalization and accelerates recovery of patients, relief of sore throat, muscle pain, runny nose, and normalization of body temperature. Kan Jang^®^/Nergecov^®^ significantly relieves the severity of inflammatory symptoms such as sore throat, runny nose, and muscle pain, decreases pro-inflammatory cytokine IL-6 level in the blood, and increases patients’ physical performance (workout) compared to placebo. In this study, for the first time we demonstrate that Kan Jang^®^/Nergecov^®^ is effective in treating mild COVID-19.

## 1. Introduction

Over 137 interventional clinical trials of antiviral treatments in Mild COVID-19 are currently in progress, and 52 studies have been completed [1]. Most of these studies are focused on the efficacy of conventional drugs, including antiviral (remdesivir), anti-inflammatory (dexamethasone), and anti-malarial (hydroxychloroquine) drugs, and show only moderate benefit along with many adverse effects. Other studies are conducted with herbal preparations, such as *Andrographis paniculata* (Burm. F.) Wall. ex. Nees extracts [2,3,4,5,6,7,8], Shen Cao Gan Jiang Tang decoction (fixed combination of *Glycyrrhiza uralensis* Fisch., *Zingiber officinale* Roscoe, and *Panax ginseng* C.A.Mey.) [9], Bronchipret sirup (fixed combination of thyme (*Thymus vulgaris* L.), ivy (*Hedera helix* L.) or cowslip (*Primula veris* L.) [10], Lian Hua Qing Wen Capsules (the major ingredients of LH consisted of *Forsythia suspensa* (Thunb.) Vahl, *Lonicera japonica* Thunb., *Ephedra sinica* Stapf, *Isatis indigotica* Fortune ex Lindl., *Pogostemon cablin* (Blanco) Benth, *Rheum palmatum* L., *Glycyrrhiza uralensis*, *Dryopteris crassirhizoma* Nakai, *Rhodiola crenulata* (Hook.f. and Thomson) H.Ohba, *Houttuynia cordata* Thunb., *Prunus sibirica* L., gypsum, and 1-menthol) [11] and natural compounds of plant origin or their derivatives such as cannabidiol [12], artemisinin/artesunate [13], resveratrol [14], curcumin and quercetin [15,16] and iso-andrographolide-19-sodium sulfate [17].

Remarkably, some of these products include adaptogenic plants (Panax Ginseng, Andrographis, Glycyrrhiza, Rhodiola species), which increase resilience and survival due to activation of repair and detoxifying processes and normalization of cellular homeostasis. It was suggested that adaptogens could help prevent and treat COVID-19 at various stages of progression and resolve the disease due to their pleiotropic effects on virus replication, the immune system, and recovery of oxidative stress-induced damage in compromised cells and tissues [18]. Recently, the clinical efficacy of Chisan^®^/ADAPT-232 (a fixed combination of Rhodiola, Schisandra, and Eleutherococcus) in ameliorating Long COVID-19 symptoms has been demonstrated [19].

However, published studies with *Andrographis paniculata* extracts, officially used in Thailand [20,21,22] for treating the acute phase of COVID-19, are contradictory [2,3,4].

A retrospective cohort open-label observational study of the efficacy of *Andrographis paniculata* (Burm.f.) Wall. ex Nees crude extract in capsules (total dose of andrographolide 180 mg/day for five consecutive days) in addition to supportive treatment (antipyretics, mucolytics, expectorants, antihistamines, oral rehydration salts, and anxiolytics) in hospitalized mild COVID-19 patients does not reveal a significant difference in the incidence rates of developing pneumonia versus supportive treatment [4].

On the contrary, two randomized, placebo-controlled studies show superior efficacy of *Andrographis paniculata* extract (APE, at the daily dose of 60 mg andrographolide, for five days) in 63 adults with mild COVID-19 vs. placebo, clinical recovery rates, pneumonia, changes of serum CRP levels, and nasopharyngeal SARS-CoV-2 detection by rRT-PCR on Day 5 [2,3].

These results are in line with the results of a prospective, multicenter, open-label, and randomized controlled trial of Xiyanping intravenous injection (iso-andrographolide-19-sodium sulfate, a synthetic derivative of andrographolide, the primary anti-inflammatory compound of Herbal Andrographidis) in patients with mild to moderate COVID-19, where significant reduction in the time to cough relief, fever resolution and virus clearance were demonstrated [17]. Furthermore, they agree with preclinical in vitro and in vivo studies of *Andrographis paniculata* and *Eleutherococcus senticosus* preparations [23,24,25,26,27,28,29,30,31,32,33,34,35,36,37,38]. Antiviral activity of *Eleutherococcus* extracts were demonstrated in experimental model systems in which rodents were infected with H1N1 influenza A virus [23,24,25,26], human rhinovirus and respiratory syncytial virus [24]. The antiviral effect of *Andrographis* extracts [27] were demonstrated against H1N1 influenza A [28,29,30], H5N1 avian influenza [31], Chikungunya [32] Dengue [33,34], and SARS-CoV-2 [35,36,37,38] viruses.

Recently, the predicted antiviral effect of *Andrographis* extracts against SARS-CoV-2 virus docking and replication by targeting Nsp5, Nsp3, Nsp12, Nsp1 structural proteins and S2 Spike glycoprotein receptor to type-II transmembrane serine protease enzymes of host cells was also demonstrated by in silico modeling [39,40,41,42,43], suggesting the potential efficacy of Andrographolide (I) and its derivatives in COVID-19.



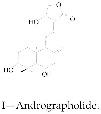



Furthermore, it has been shown that *Andrographis* is more active in combination with *Eleutherococcus* in preventing the growth of SARS-CoV-2 in isolated Vero E6 human primary embryonic kidney epithelial cells [38]. The synergistic interaction of ES and AP was demonstrated in the expression of genes involved in inflammatory and immune responses [44]. The synergistic effect is assumed to be associated with their impact on neuroprotection, tumor cell proliferation, regulation of Nrf2-mediated signaling proteins, and enzymes related to antioxidants, metabolization, and detoxification [45,46].

Both adaptogenic plants, *Andrographis paniculata* (Burm. F.) Wall. ex. Nees, *Acanthaceae* (AP), *Eleutherococcus senticosus* (Rupr. & Maxim.) Maxim, *Araliaceae* (ES), and their fixed combination Kan Jang are known for antiviral, immunomodulatory, and anti-inflammatory activity [18] and clinical efficacy in the respiratory tract of patients with infectious diseases [47,48,49,50,51,52,53,54,55,56,57]. Kan Jang^®^ is a registered herbal medicinal product in Denmark (marketing authorization Nr 6132293) with well-established use for alleviating symptoms of viral respiratory diseases, such as common colds and influenza, since 1997.

This study aimed to assess the clinical efficacy and safety of Kan Jang^®^, in 140 patients with mild and moderate COVID-19 inflammatory symptoms [58], in the last three days) and not requiring Intensive Care Unit (ICU) admission. However, after recruiting 86 patients, the limitation of 3 days has been amended for seven days starting from 2022-05-06 due to the spread of the SARS-CoV-2 Omicron variant in Georgia and the lack of patients admitting into the hospital with mild COVID-19 symptoms within three days of the manifestations of the symptoms. Therefore, this article provides the interim analysis results in the subgroup of patients with mild and moderate COVID-19 inflammatory symptoms in the last three days.

## 2. Results

### 2.1. Patients

#### Demographic and Baseline Characteristics

Nighty eight patients with confirmed diagnoses based on a positive SARS-CoV-2 test, experiencing mild to moderate COVID-19 symptoms, were assessed for eligibility. Eighty six patients with at least three of eight COVID symptoms (fatigue, headache, nasal discharge, loss of smell, taste, cough, muscle pain, and body temperature from 37 to 38 °C) for the last three days before admission to the hospital, were randomly assigned to two treatment groups, Kan Jang^®^, (A) and Placebo (B), Figure 1.

The groups did not show differences in baseline demographic, physical, and other critical clinical measurements, Table 1, carried out during the study or identified as important indicators of prognosis or response to therapy characteristics.

### 2.2. Efficacy

#### 2.2.1. Primary Endpoints

The primary efficacy endpoints were: (i) the rate of patients (%) with clinical deterioration, (ii) the duration of hospitalization, (iii) the time to virus clearance, and (iv) the duration of the acute phase of disease assessed as the time from the start of study medicine to complete symptom resolution, (v) fever resolution and relief the severity of fatigue, headache, sore throat, cough, rhinorrhea (nasal discharge/runny nose), myalgia (muscle pain), loss of smell and taste.

##### The Rate of Patients with Clinical Deterioration and Duration of Hospitalization in the Treatment Group and Control Group

In Kan Jang^®^, Group A of 34 patients, four were dropouts, and three were withdrawn from the study due to lack of efficacy and disease progression; they continued the treatment with steroids and antibiotics. Figure 1.

In Placebo Group B, of 52 patients, 11 were dropouts, and ten were withdrawn from the study due to lack of efficacy and disease progression; they continued the treatment with steroids and antibiotics. Figure 1.

Therefore:Group A’s (Kan Jang, 30 patients) disease progression rate (10.0%).Group B’s (placebo, 41 patients) disease progression rate (24.39%).

Where *p* < 0.05 (significant result) at 90% significance level (Confidence 90%), power 78.9%. Figure 2a.

The primary endpoint was met since Kan Jang^®^ significantly reduced disease progression rate compared to placebo. The disease progression rate in the placebo group was 24.39%, which is 143.9% higher than the disease progression rate in the Kan Jang Group (10.00%), Figure 2. Absolute risk reduction by Kan Jang treatment is 14% (ARR = 0.14), while the relative risk reduction (RRR, %) = 243.9%. The number of patients required to treat with Kan Jang to prevent one additional bad outcome (defined as the Number Needed to Treat, NNT = 1/ARR = 7.14) comprises 7.14.

##### Virus Clearance and Body Temperature

The percentage of patients with a negative SARS-CoV-2 virus test was 14% lower in the Kan Jang group compared to placebo after seven days (76.47% vs. 90.38%, difference −13.91 ± 8.5%, *p* = 0.89) and 14 days (35.05% vs. 48.86%, difference −13.81 ± 9.7%; *p* =0.90) of the treatment, Figure 3a. The rate of virus clearance was faster, and the time to virus clearance in 50% of patients was 2.6 days shorter in the Kan Jang group (11.4 days) compared to placebo (14 days); hazard ratio Kan Jang/placebo = 1.780, 95% CI of ratio from 0.7382 to 4.291, Figure 3a.

Duration of increased body temperature (from >37 °C to <38 °C) was also shorter in the Kan Jang group vs. placebo group; median recovery in Kan Jang^®^ was seven days, while in placebo it was nine days; hazard ratio Kan Jang/placebo = 1.125, 95% CI of ratio from 0.5778 to 2.191 (Figure 3b).

##### The Duration and Severity of Inflammatory Symptoms

The occurrence of various inflammatory symptoms was quite different at the baseline of the cohort of COVID-19 patients recruited in this study (Table 1); the most common symptoms were fatigue (in 100% of patients), headache (in 85% of patients), sore throat (in 42% of patients), muscle pain and cough (in 38% of patients), while rhinorrhea and loss of smell were observed correspondingly in 12% and 8% of patients. No patient with loss of taste was recorded at the beginning of the study.

The severity of all inflammatory symptoms gradually decreased from the baseline to the end of therapy (Day 14) and the follow-up period (21 days) in both groups of patients. However, a significant improvement over time of treatment and the follow-up period in the Kan Jan group compared to placebo was observed in relief of the severity of sore throat, cough, myalgia (muscle pain), and rhinorrhea (nasal discharge/runny nose), myalgia, Figure 4, Figure 5 and Figure 6.

The therapeutic efficacy of Kan Jang^®^ was assessed by comparing (i) differences in the time to resolve inflammatory symptoms (Figure 4a, Figure 5a, Figure 6a and Appendix A) in the Kan Jang^®^ and placebo groups of patients and (ii) differences in the relief in severity of inflammatory symptoms from the baseline (Figure 4b, Figure 5b, Figure 6b and Appendix A) in the Kan Jang^®^ and placebo patient groups. Treatment groups were compared for all efficacy outcome measures to assess primary and secondary endpoints.

Assessment of the efficacy of Kan Jang treatment on the severity of overall inflammatory symptoms, including runny nose, plugged nose, sneezing, sore throat, cough, hoarseness, head congestion, chest congestion, and feeling tired, using Wisconsin URS subway, did not reveal a significant difference compared to the placebo effect, despite considerable improvement in both groups over the time of treatment and the follow-up period, Appendix A.

The rate of reduction in cough in the Kan Jang group was insignificantly higher than in the placebo group, Figure 7a; e.g., the percent of patients with cough in the Kan Jang group was 13% less than in the placebo group, and the median recovery in Kan Jang^®^ was nine days, while in placebo it was eleven days; hazard ratio Kan Jang/placebo = 1.345, 95% CI of ratio from 0.4683 to 3.863, Figure 7a. Surprisingly, the decrease in the severity of the cough was insignificantly smaller in Kan Jang vs. placebo, Figure 7b.

#### 2.2.2. Secondary Endpoints

Secondary endpoints comprised the measures of (i) immune response marker IL-6 concentration in the serum, (ii) blood hypercoagulation marker Dimer-D, (iii) inflammatory marker C-reactive protein, (iv) physical activity, (v) physical performance, (vi) cognitive performance, and (vii) severity of respiratory symptoms and quality of life by Wisconsin Upper Respiratory Symptom Survey Questionary Score.

##### Blood Serum Markers of Immune Response and Inflammation

At the beginning of the study, the baseline level of all selected markers of immune response and inflammation, IL-6, C-reactive protein, and D-dimer, were significantly higher than normal blood values, Table 1. In 7 days of the treatment with Kan Jang, the level of IL-6 reached the standard limit of 7 pg/mL, while in the placebo group, two days longer, Figure 8a. Between-groups comparison of the changes of level of cytokine IL-6 in the blood from the baseline over time shows significant interaction (*p* = 0.0186), Figure 8b. The Kan Jang^®^ treatment has a statistically significant (*p* < 0.0001) effect on cytokine IL-6 in blood compared to the placebo.

The level of C-reactive protein and D-dimer was also normalized in the blood of all patients at the end of treatment; however, the Kan Jang^®^ treatment had no statistically significant difference in effects on C-reactive protein and D-dimer compared to placebo *p* > 0.05, Appendix A.

##### Physical Activity, Physical and Cognitive Performance, and Quality of Life Scores

The Kan Jang^®^ treatment significantly increases patients’ physical performance/workout time (in min) compared to the placebo, Figure 9a; between-groups comparison of the changes from the baseline over time of treatment and follow-up period shows significant interaction (*p =* 0.0173), Figure 9a. However, subjective assessment of overall physical activity (Figure 9b) and cognitive performance scores (Appendix A) using self-assessment questionaries did not provide a statistically significant difference between Kan Jang^®^ and placebo groups.

### 2.3. Safety

No adverse events were noted in the study. Regardless of causality, adverse events were monitored for all patients from the first dose and through the one-week follow-up period.

## 3. Discussion

Recently it was suggested that adaptogens could be useful in prophylaxis and treatment of viral infections, including SARS-CoV-2 infection at all stages of progression of inflammation [18], including the acute phase of the disease and the chronic post COVID period (long COVID, usually for three months from the onset of COVID-19 with symptoms that last for at least two months, mainly fatigue, shortness of breath, cognitive dysfunction, etc.). Indeed, recently, it has been demonstrated that a fixed combination of Rhodiola, Schisandra, and *Eleutherococcus* (Chisan/ADAPT-232) can increase physical performance and reduce the duration of fatigue and chronic pain in patients with long COVID-19 symptoms. Chisan^®^ also reduced the blood level of pro-inflammatory markers IL-6, C-reactive protein, and creatinine, suggesting prevention of renal failure progression in long COVID [19]. Another fixed combination of adaptogens (*Andrographis* and *Eleutherococcus*, Kan Jang) was studied in the acute phase of COVID-19. In this study, we, for the first time, show that Kan Jang effectively relieves sore throat, muscle pain, and runny nose in COVID-19, Figure 4, Figure 5 and Figure 6. This is in line with previous studies of Kan Jang in other upper respiratory tract infections, such as common cold, rhinitis, sinusitis [47,51,54,55], and influenza [53].

This effect is associated with the anti-inflammatory effect of Kan Jang on the pro-inflammatory cytokine IL-6, which is decreasing to normal levels in the blood of patients within seven days of treatment (Figure 7), presumably with faster virus clearance, Figure 3a. Additionally, Kan Jang significantly improves patients’ physical performance (workout), increasing the time of walking during the first week of recovery. Figure 8.

Furthermore, the duration of increased body temperature (Figure 3b) and hospitalization (Figure 2b) decreased in the Kan Jang group of patients vs. placebo.

Finally, the rate of patients with clinical deterioration and disease progression was significantly lower in the Kan Jang group vs. the placebo, Figure 2a. The disease progression rate in the placebo group was 24.39%, which is about 2.5-fold higher than the disease progression rate in the Kan Jang Group (10.00%), Figure 2. Absolute risk reduction by Kan Jang treatment is 14% (ARR = 0.14), while the relative risk reduction (RRR, %) = 243.9%. The number of patients required to treat with Kan Jang to prevent one additional bad outcome (defined as the Number Needed to Treat, NNT = 1/ARR = 7.14) comprises 7.14. The higher ARR and lower NNT are, the more effective the intervention is.

Unlike Xiyanping injection (iso-andrographolide-19-sodium sulfate) [17] or Lian Hua Qing Wen herbal combination [11], Kan Jang treatment does not relieve cough and fatigue in 32 patients in our study. However, it cannot be excluded that these effects will be observed in the larger sample size at the end of our study.

The treatment with Kan Jang^®^ was well tolerated, and no adverse events were recorded in this study. This is in line with previous clinical studies where only four adverse events were recorded in one [47] of six studies [47,51,53,54,55,57] with 494 patients in the Kan Jang group. In that study, the only common adverse reaction was mild pruritus observed in four patients in the Kan Jang^®^ group and six in the placebo group (odds ratio 5852, z-statistics 0.807, *p* = 0.4194), and no severe adverse reactions were observed.

It comprises 0.76% of the seven studies’ 526 patients in the Kan Jang^®^ groups. That is significantly less than 131 mild to moderate adverse events observed in 690 (19%) patients of ten studies of *Andrographis paniculata* herbal preparations that are essentially safe [59], and 383 (4.04%) adverse events in 9490 participants using andrographolide derivative injections, which can be life-threatening, mainly gastrointestinal, skin and subcutaneous tissue disorders, and anaphylaxis [59].

Kan Jang has an excellent safety profile [47,48], presumably due to the adaptogenic and antitoxic effects of *Eleutherococcus* [60]. Cytoprotective (neuroprotective, hepatoprotective, cardioprotective), stress-protective, antioxidant, antitoxic and immunomodulating activity of *Eleutherococcus* preparations were demonstrated in many experimental models on isolated cells and (in vitro and ex vivo) animals [60,61,62].

This evidence of the safety of the Kan Jang combination shows that pharmacological activity and toxicity of multi-component drugs are different from their ingredients, and the effects observed on purified compounds cannot be extrapolated to their combinations with other substances and vice versa, due to their multiple synergistic and antagonistic interactions in the organism [44,45,46]. In other words, the results obtained in this study are product specific and cannot be applied to mono herbal drugs, e.g., *Andrographis* extract, which in turn is the mixture of 39 compounds such as andrographolides, flavonoids, etc. and deregulates quite a different number of genes than expected from a simple calculation of a number of constituents of the plant extracts [44], Table 2.

Currently, only a few antiviral drugs are approved by drug regulatory authorities against COVID-19: Paxlovid™ [63], Remdesivir, Bebtelovimab (in USA) [64] Molnupiravir (in UK) [65], and *Andrographis* (in Thailand) [20,21,22] for treating non-hospitalized COVID-19 patients. The FDA has also approved the immune modulators for certain hospitalized adults with COVID-19, e.g., Olumiant (baricitinib), Interleukin-6 receptor blocker [66].

In this context, an important difference in the mode and mechanisms of action of Kan-Jang and *Andrographis* compared with other drugs, is its multitarget effects both directly on virus-receptor binding, viral membrane fusion into the host cell, viral replication, transcription, translocation, assembly, and release to infect other host cells [18,67], as well as indirect antiviral activity due to multiple effects on innate and adaptive immune systems, inflammatory response, and essentially on recovery of oxidative stress-induced damage in compromised cells [18].

Overall, the safety of Kan Jang is an essential advantage compared to the safety of all other anti-COVID-19 drugs [63,68,69]. For instance, the recently approved Paxlovid™ (Pfizer), a co-packaged combination of nirmatrelvir and ritonavir tablets, which effectively reduces the risk of progression (ARR of 6.2%, NNT = 17) to severe COVID-19 in symptomatic adults at high risk of progression to severe COVID-19 [59]. That is, the safety of Kan Jang compared to Paxlovid™, which, unlike Kan Jang, induces dysgeusia, diarrhea, hypertension, and myalgia [63].

The limitations of our study are a short duration of symptoms of COVID-19 before treatment (for three days), and lack of concomitant chronic diseases, which increase the risk of progression of disease and intensive care therapy for patients. Further studies are required in patients with a more extended pretreatment period of COVID-19 symptoms (up to 7 days) and patients at high risk of progression of pneumonia, acute respiratory distress syndrome, and septic shock.

## 4. Materials and Methods

### 4.1. Study Design, Recruitment, and Screening of Patients, Schedule of Examinations

This prospective, randomized, placebo-controlled, quadruple-blind, two-parallel-group (Figure 1 and Appendix A), phase II interventional study was conducted at the Tbilisi State Medical University, Tbilisi, Georgia, with the approval of the Biomedical Research Ethics Committee of Tbilisi State Medical University and National Council on Bioethics (Registration Nr 3-2021/87, date of final protocol approval 25 March 2021). ClinicalTrials.gov Identifier: NCT04847518. https://www.clinicaltrials.gov/ct2/show/NCT04847518 (accessed on 6 June 2022).

Recruitment for the study was initiated on 24 April 2021, and the 86th patient was recruited on 22 October 2021.

All essential principles of the declaration of Helsinki, the ICH guidelines, and EMEA clinical trials guidelines were considered. In the course of the initial visit to the University Clinic, patients’ exclusion and inclusion criteria were verified against the eligibility checklist, and patients interested in participating received relevant information about the study. All patients provided written informed consent to enter the study before inclusion. The patients had enough time to consider the information, to confirm that they understood it and were willing to participate in the study. The patient’s information sheet described the study procedures, the aims, expected benefits, and potential risks in Georgian and English. Patients were evaluated by a physical examination, and suitable lab tests were conducted. Patients underwent randomization when inclusion criteria were met.

Overall, 98 patients were assessed for eligibility, and a subgroup of 86 patients with mild COVID symptoms [45] for the last three days, were randomized and included in the interim analysis of the study (Appendix A, Full Analysis Dataset), which is still in progress. In total, 69 patients (80% randomized) completed their respective treatment cycles according to protocol, while 17 (20%) discontinued therapy after receiving at least one dose of study preparations due to the patient’s request. Fifty-eight patients, who completed the treatment, were evaluated for treatment efficacy for two weeks (visits 2 and 3) of treatment and one week after completing the treatment (follow-up visit 4) and compiled. Primary efficacy subset: 15 patients were lost in the first week of treatment. The distribution between the study groups and the disposition of patients are shown in the flow chart in Figure 1. The schedule of procedures and examinations is shown in Table 3.

#### 4.1.1. Study Population, Inclusion, and Exclusion Criteria

The population for this study aged 18 years and older of either sex (mean age: 48.85 ± 13.86 years) consisted of COVID-19 patients in stable, moderate condition (i.e., not requiring Intensive Care Unit (ICU) admission) with confirmed diagnosis based on positive SARS-CoV-2 test and at least three of eight mild to moderate COVID-19 symptoms [58], such as fatigue, headache, sore throat, nasal discharge, cough, pain in muscles, loss of taste and smell (Table 1) for the last three days before recruitment for the study. Subjects must be under observation in the hospital, be enabled to take medication alone, and give written informed consent.

Patients admitted with severe acute respiratory syndrome under invasive mechanical ventilation and diagnosed with an etiologic agent other than COVID 19 were excluded from the study. Other exclusion criteria were: acute and chronic pulmonary diseases, chronic rhinosinusitis, renal failure or creatinine ≥2.0 mg/dL, type 2 diabetes, autoimmune disease, patients taking antibiotics for a reason other than COVID-19 at enrollment, chronically suppressed immune system (AIDS, lymphoma, corticosteroid therapy, chemo-or radiotherapy for last six months), cancer patients taking immunosuppressive drugs, pregnant or lactating women, subjects participating in other clinical studies or taking any medication influencing the outcome measures during the clinical trial.

#### 4.1.2. Participant Withdrawal

According to the study protocol, the participants should be withdrawn if they develop an allergy or hypersensitivity to study preparation, in case serious side effects appear, in case abnormal clinical biochemistry values are found, and in case of subjects who do not comply with the study protocol. Patients were free to withdraw from the clinical study at any time without giving a reason.

#### 4.1.3. Data Sets Analyzed

All enrolled and randomly allocated treatment patients were included in the intention to treat (ITT) analysis. Efficacy subset analysis per protocol (PP) was performed for the subset of patients with mild COVID symptoms at the baseline (visit 1) and when they completed the study therapy (visits 3 and 4). A per-protocol (PP) analysis aims to identify a treatment effect on the symptoms. Therefore, some patients were excluded from the complete analysis set (ITT), and the PP population was used for the PP analysis, Figure 1.

### 4.2. Intervention and Comparator

Pharmaceutical-grade standardized extracts of *Andrographis paniculata* L. Nees. (herb) and *Eleutherococcus senticosus* (Rupr. & Maxim.) Maxim (root) genuine extracts, as well as their fixed combination, Kan Jang^®^, were manufactured, tested, and then released for human use as per ICH Q7A and EMEA guidelines for Good Agricultural and Collecting Practice and Good Manufacturing Practice (GMP) of active pharmaceutical ingredients at the Swedish Herbal Institute, which holds a valid EU-GMP license to produce pharmaceuticals.

One capsule of Kan Jang^®^/Nergecov^®^ (size 0, batch No. 40154, Expiry date: February 2024) contains 260 mg of *A. paniculata* native extract SHA-10 (drug-native extract ratio of 4.5–8.1: 1, extraction solvent 70% ethanol), including 15 mg of diterpene lactones (andrographolide and 14-deoxy-11,12-didehydroandrograholide), and 18.75 mg of *E. senticosus* native extract (drug-native extract ratio of 17–30: 1, extraction solvent 70% ethanol, 0.25 mg Eleutherosides B and E), for details see Appendix A. The matrix contained inactive excipients (microcrystalline cellulose and magnesium stearate). The placebo capsules (size 0, batch No 40154) containing the inactive excipients were identical to the Kan Jang^®^ capsules. Both preparations’ appearance, smell, and color were similar and organoleptically undistinguishable.

The investigational products (IPs) were packaged, blinded, labeled, and packed by the Swedish Herbal Institute AB, Sweden, according to national requirements regarding the use for clinical trial investigation. The label included the drug name, study code, and storage conditions. Placebo and Kan Jang^®^ packages were provided to patients in packages containing 90 capsules in six blisters (the amount required for the treatment period of 14 days is 84 capsules). Each patient received packages on visit 1 (Day 1). The principal investigator (PR) recorded the participant’s ID (the name of the patient allocated to the treatment code number) on the label and provided that package to the patients at the visit 1.

Herbal preparation was qualitatively and quantitatively analyzed according to product specifications. All analytical methods were validated for accuracy, precision, and selectivity. Reference samples were retained at the Swedish Herbal Institute AB (Vallberga, Sweden).

The investigational product was accessible only to authorized personnel and maintained under controlled storage conditions specified on the label and in the investigator’s brochure.

#### 4.2.1. Doses and Treatment Regimens

The daily dose of the study intervention was two capsules three times per day for two consecutive weeks with a daily intake of 1560 mg of the dry extract *A. paniculata* SHA-10, corresponding to 90 mg of diterpene lactones andrographolides from *Andrographis paniculata* herb and 112.5 mg of dry extract of *E. senticosus*, in the Kan Jang^®^ group. All the patients were provided with diary cards on which the daily consumption of study preparation was recorded. The number of consumed capsules during treatment by every single patient was verified for compliance with the duration of the treatment/illness (days at the clinic). The principal investigator was responsible for maintaining drug accountability records.

#### 4.2.2. Randomization and Blinding

Study preparations were labeled by a quality assurance responsible person (QP) at the manufacturing site. The QP did use a random numbers sequence, which was generated by PRISM GraphPad software (2017 Online version, GraphPad Software Inc., San Diego, CA, USA) “Random number generator” (https://www.graphpad.com/quickcalcs/randomize1.cfm, accessed on 16 February 2022). The randomization sequence comprises a table of two columns (A and B) filled with randomly distributed unique numbers from 1 to 160. It provided the information on the content of each package—how placebo and verum packages/containers were labeled. QP as-signed the codes A and B to Kan Jang and placebo sets of packages.

#### 4.2.3. Allocation Concealment

All the packages have allocation concealed random numbers printed on the label. The randomization sequence list and code were blinded from the PI, so the PI assigned the packages sequentially as per numbers. The random sequence was disclosed to the statistician before the statistical evaluation of the results when 86 patients had completed the treatment.

#### 4.2.4. Implementation and Blinding

At the first visit, participants received a consecutive number starting from 1 to 86. They were linked to a unique number according to the randomization sequence. Patients were sequentially enrolled by the PI and assigned to a random number, and received the capsules in the package. The investigator produced the participant list and gave the treatment code number (from 1 to 86) to every single patient. He recorded the patients’ names in case report forms (CRF) and on the labels of the package. The table shows the names of patients and corresponding study preparation numbers (treatment number mentioned on the label of packages).

Blinding for trial subjects was achieved using labeled packages containing capsules of the same appearance. The study product was delivered to the clinic pre-labeled and coded according to the randomization list.

The study participants’ list, identifying the patients and the investigational product packages (numbers), were kept by the principal investigator. The treatment code providing the information about the actual assignment of groups A and B to Kan Jang^®^ and placebo were broken by QP, and after that, a statistical analysis of the datasets was completed, and the results of the study were obtained. Thus, the study was quadruple-blind since study preparations were blinded to all investigators, care providers, participants, and outcome assessors.

#### 4.2.5. Evaluation of Compliance

Patients were asked to take their daily dose of 6 capsules. They were questioned about their overall compliance with the study protocol upon their visits, and the study personnel measured the remaining capsules. He verified patients’ records in a unique form enclosed in the case report form. The doctor monitored overall compliance with the study protocol upon their visits and counted the remaining capsules at the treatment.

### 4.3. Efficacy and Safety Outcomes and Endpoints

The efficacy endpoints were the differences in duration and the relief of inflammatory symptoms in Kang Jang and placebo groups of patients. The outcomes include changes in the severity of inflammatory symptoms, measured by means of various scales’ scores, from the baseline to the end of therapy (Day 14) and follow-up period (21 days after randomization).

The primary efficacy outcome measures of the study were: (i) the rate of patients (%) with clinical deterioration, (ii) duration of hospitalization, (iii) the time to virus clearance, (iv) the duration of the acute phase of disease assessed as the time from the start of study medicine to complete symptom resolution, (v) fever resolution and relief, the severity of fatigue, headache, sore throat, cough, rhinorrhea (nasal discharge/runny nose), myalgia (muscle pain), loss of smell and taste.

Secondary endpoints comprised the measures of (i) Immune response marker IL-6 concentration in the serum, (ii) blood hypercoagulation marker Dimer-D, (iii) inflammatory marker C-reactive protein, (iv) physical activity (v) physical performance, (vi) cognitive performance, and (vii) severity of respiratory symptoms and quality of life by Wisconsin upper respiratory symptom survey questionary score.

Safety and tolerability were assessed by monitoring the incidence and duration of adverse events.

### 4.4. Statistical Analysis

All the clinical data were recorded in standardized case report forms and tabulated in an Excel dataset (Appendix A), which was used in statistical analysis by Prism software (version 3.03 for Windows; GraphPad, San Diego, CA, USA).

Statistical analysis was performed using “observed” data for time-to-event outcomes of the intent-to-treat (ITT) population, defined as all randomly assigned participants who received at least one dose of the study product.

The mean outcomes were evaluated at baseline for patients who received Kang Jang vs. placebo by the Student’s parametric independent-measures *t*-test (variables with normal distribution) or Mann–Whitney non-parametric test, depending on the results of the D’Agostino and Pearson omnibus normality test. Within-group repeated measures analysis of variables was conducted with one-way ANOVA (data with normal distribution) or the Friedman non-parametric rank test.

For assessment of the duration of the symptoms, Kaplan–Meier curves were generated for all endpoints, and medians were calculated from those curves. The treatment arms were compared by Gehan–Breslow–Wilcoxon and Mantel–Cox log-rank tests depending on the results of the normality test. The estimates of treatment hazard ratios based on log-rank tests and 95% CIs were calculated.

Evaluation of the efficacy of study preparation was achieved by comparison of mean changes from the baseline (differences before and after treatment of particular participant) between groups using two-way between–within ANOVA in which an interaction effect indicates a different response over time between the two groups and would therefore indicate a treatment effect, as well as by multiple comparison *t*-test (one unpaired test per row). The level of statistical significance was set at 5% in all methods.

#### Sample Size Considerations

Assuming that the standardized difference of means between groups for the symptoms is 0.6 (this assumption is made on the results of previous studies [47,51,53,55,57] where target differences and SD were estimated), and the power of 80% is acceptable to detect this difference as statistically significant at the 5% level, the sample size of 86 patients (42 patients in each of two groups) was calculated using monograms while comparing sample size and power for two treatment groups. Additionally, the sample size was calculated using the formula *n* = 16σ2/W2, where *n* is the minimal sample size, σ is the variance, and W is the confidence interval size. This method suggested a minimal sample size of 28. The necessary sample size was estimated at 80, i.e., 40 subjects per arm. Predicting a 30% dropout rate, which would increase our intended sample size, 70 patients in each group would be enough to detect a statistically significant difference between study groups. The proposed sample size (*n =* 140, or 70 per treatment condition) has 90% power to detect an effect size of 0.76 and 80% power to detect an effect size of 0.85, using a 2-group *t*-test with a 0.05 two-sided significance level.

## 5. Conclusions

This study provides new evidence on the clinical efficacy and safety of adaptogens, specifically Kan Jang^®^/Nergecov^®^, in the acute phase of mild COVID-19. Kan Jang^®^/Nergecov^®^ reduces the risk of progression of disease, significantly reduces the duration of disease, virus clearance, and days of hospitalization, and accelerates recovery of patients, relief of sore throat, muscle pain, runny nose, and normalization of body temperature. Kan Jang^®^/Nergecov^®^ significantly relieves the severity of inflammatory symptoms such as sore throat, runny nose, and muscle pain decreases pro-inflammatory cytokine IL-6 level in the blood, and increases patients’ physical performance (workout) compared to placebo.

## Figures and Tables

**Figure 1 pharmaceuticals-15-01013-f001:**
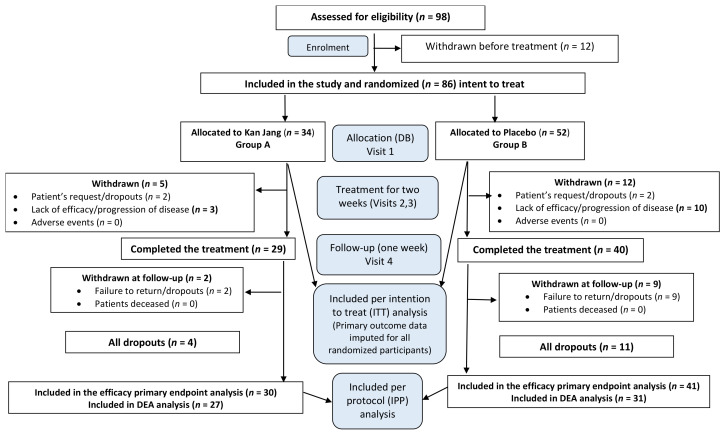
Schematic diagram of the trial. For details on the disposition of patients, see Appendix A.

**Figure 2 pharmaceuticals-15-01013-f002:**
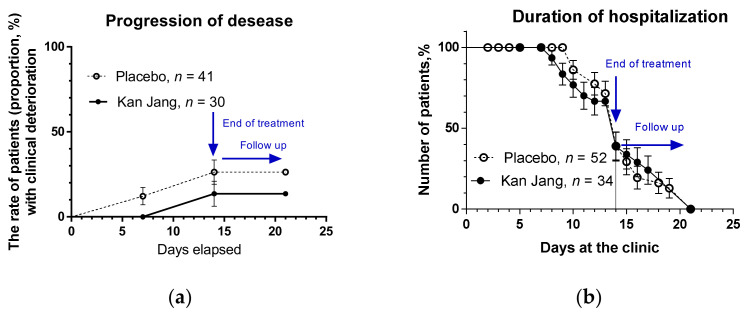
(**a**) The rate of patients with clinical deterioration in the treatment and control groups; hazard ratio Kan Jang/placebo = 0.4234, 95% CI of ratio from 0.132 to 1.357. (**b**) Duration of hospitalization in the treatment group and control group; Kaplan–Meier curves show the percent of patients hospitalized over the time from randomization (Day 1) to the end of the treatment (Day 14) and followed up for one week (Day 21) in the treatment and control groups; hazard ratio Kan Jang/placebo = 0.9398, 95% CI of ratio from 0.4978 to 1.774.

**Figure 3 pharmaceuticals-15-01013-f003:**
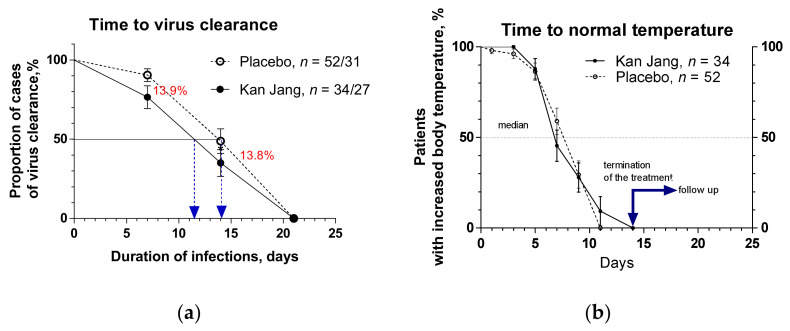
(**a**) The virus clearance in the treatment and control groups: Kaplan–Meier curves show the percent of patients with SARS-CoV-2 virus over the time from randomization (Day 1) to the end of the treatment (Day 14) and the follow-up period for one week (Day 21) in the treatment and control groups; hazard ratio Kan Jang/placebo = 1.891, 95% CI of ratio from 0.5969 to 1.675. (**b**) Duration of increased body temperature (from >37 °C to <38 °C) in the treatment and control groups; median recovery: Kan Jang^®^—7 days, placebo—9 days; hazard ratio Kan Jang/placebo = 1.125, 95% CI of ratio from 0.5778 to 2.191.

**Figure 4 pharmaceuticals-15-01013-f004:**
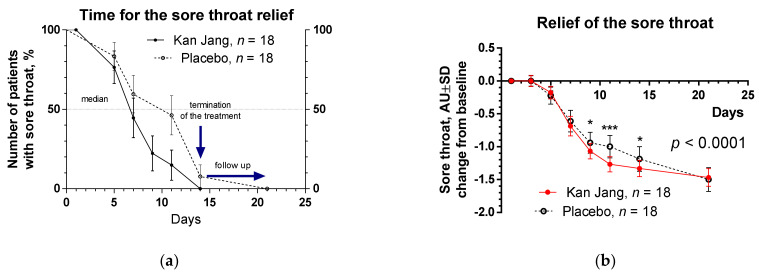
(**a**) Time to relieve sore throat in the treatment and control groups: Kaplan–Meier curves show the percent of patients with a sore throat over the time from randomization (Day 1) to the end of the treatment (Day 14) and follow up for one week (Day 21); median recovery, Kan Jang^®^ was 7 days, placebo was 11 days; hazard ratio Kan Jang/placebo = 2.427, 95% CI of ratio from 0.9352 to 6.296. (**b**) Relief of the sore throat; the changes in the severity of the symptom from the baseline of patients in group A (Kan Jang) and group B (placebo) over the time from Day 1 to Day 21. Between-groups comparison of the changes in the severity of the symptom from the baseline over time shows significant interaction (*p* < 0.0001). The Kan Jang^®^ treatment has a statistically significant effect on the relief of the sore throat compared to the placebo. * *p* < 0.05, *** *p* < 0.0001.

**Figure 5 pharmaceuticals-15-01013-f005:**
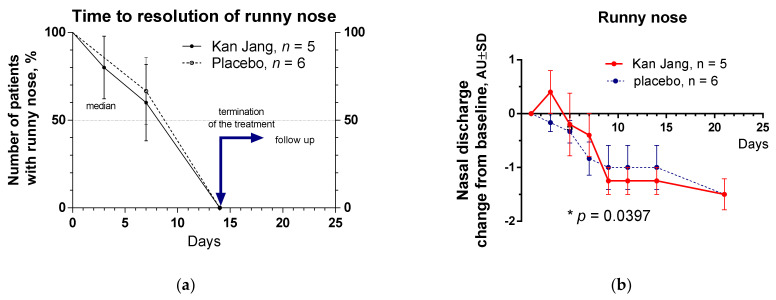
(**a**) Time to resolution of runny nose in the treatment and control groups: Kaplan–Meier curves show the percent of patients with runny nose over the time from randomization (Day 1) to the end of the treatment (Day 14) and follow up for one week (Day 21) and in the treatment and control groups; median recovery: Kan Jang^®^, was 14 days, placebo was 14 days; hazard ratio Kan Jang/placebo = 1.534, 95% CI of ratio from 0.17 to 13.57. (**b**) Reduction in nasal discharge; the changes in the severity of the symptom from the baseline of patients in group A (Kan Jang) and group B (placebo) over the time from Day 1 to Day 21. Between-groups comparison of the changes in the severity of the symptom from the baseline over time shows significant interaction (*p* = 0.0397). The Kan Jang^®^ treatment has a statistically significant effect on the reduction in nasal discharge compared to the placebo. * *p* < 0.05.

**Figure 6 pharmaceuticals-15-01013-f006:**
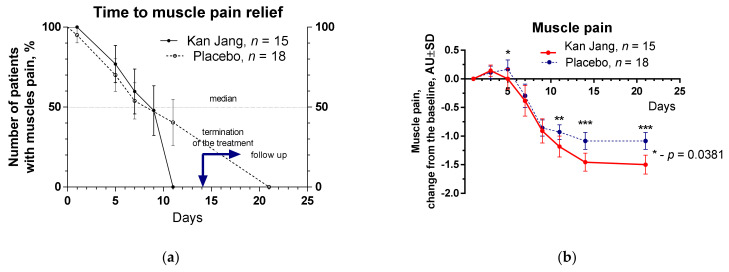
(**a**) Time to muscle pain relief in the treatment and control groups. Kaplan–Meier curves show the percent of patients with the muscle pain over the time from randomization (Day 1) to the end of the treatment (Day 14) and follow-up for one week (Day 21); median recovery, Kan Jang^®^ was 9 days, placebo was 11 days; hazard ratio Kan Jang/placebo = 1.345, 95% CI of ratio from 0.4683 to 3.863. (**b**) Relief of the muscle pain; the changes in the severity of the symptom from the baseline of patients in group A (Kan Jang) and group B (placebo) over the time from Day 1 to Day 21. Between-groups comparison of the changes in the severity of the symptom from the baseline over time shows significant interaction (*p* < 0.0001). The Kan Jang^®^ treatment has a statistically significant effect on muscle pain relief compared to the placebo. * *p* < 0.05, ** *p* < 0.001, *** *p* < 0.0001.

**Figure 7 pharmaceuticals-15-01013-f007:**
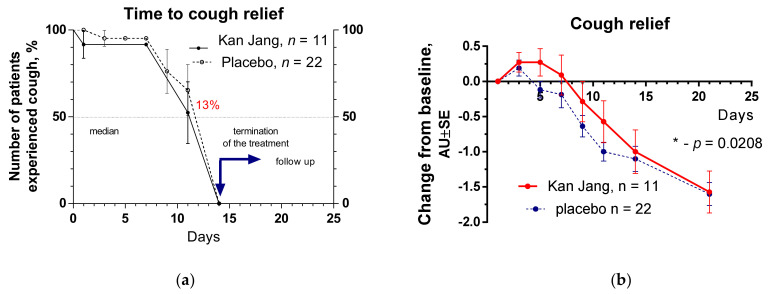
(**a**) Time to cough relief in the treatment and control groups. Kaplan–Meier curves show the percent of patients with muscle pain over the time from randomization (Day 1) to the end of the treatment (Day 14) and follow-up for one week (Day 21); median recovery: Kan Jang^®^ was 9 days, placebo was 11 days; hazard ratio Kan Jang/placebo = 1.345, 95% CI of ratio from 0.4683 to 3.863. (**b**) The changes in the severity of the cough from the baseline of patients in group A (Kan Jang) and group B (placebo) over the time from Day 1 to Day 21. Between-groups comparison of the changes in the severity of the symptom from the baseline over time shows significant interaction (*p* < 0.0001). The Kan Jang^®^ treatment has a statistically significant effect on cough compared to the placebo. *—*p* < 0.05.

**Figure 8 pharmaceuticals-15-01013-f008:**
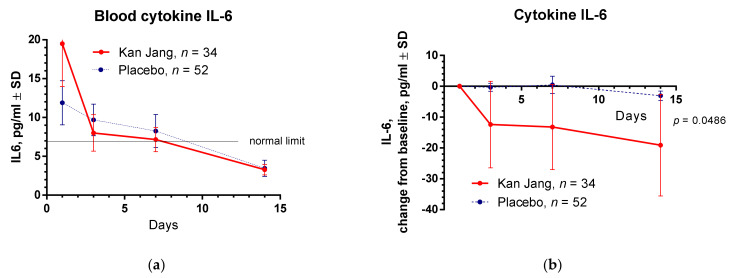
(**a**) Concentration of IL-6 (mean ± SD) in the blood of patients in group A (Kan Jang) and group B (placebo) over the time from Day 1 to Day 14. (**b**) The changes from the baseline of the levels (mean ± SD) of cytokine IL-6 in the blood of patients in group A (Kan Jang) and group B (placebo) over the time from Day 1 to Day 14. Between-groups comparison of the changes in the level of cytokine IL-6 in the blood from the baseline over time shows a significant difference (*p* = 0.0486) between groups A and B. The Kan Jang^®^ treatment has a statistically significant effect on cytokine IL-6 in blood compared to the placebo.

**Figure 9 pharmaceuticals-15-01013-f009:**
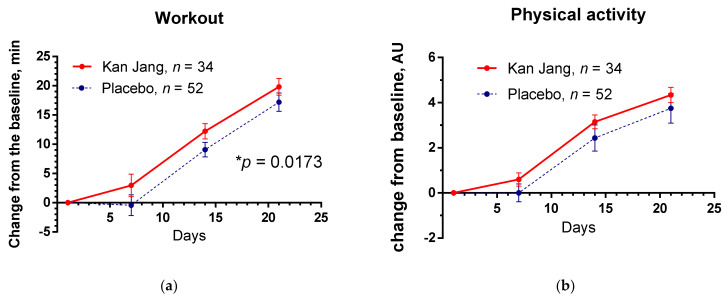
Between-groups comparison of the changes from the baseline of (**a**) physical performance/workout time (in min) and (**b**) the overall physical activity of patients in group A (Kan Jang) and group B (placebo) over the time from Day 1 to Day 21. * *p* < 0.05.

**Table 1 pharmaceuticals-15-01013-t001:** Baseline demographic characteristics, outcome measures, and laboratory biochemical and hematological measurements.

		Unit		Group A Kan Jang ADAPT *n* = 50			Group B Placebo		Signif. of Difference
Parameters			*n*	Mean	SD.	*n*	Mean	SD.	*p*-Value
Age		years	34	49.82	16.33	52	44.73	16.85	0.170
Gender		Male/Female	34	12/22 (0.54)		52	24/28 (0.86)		0.156
BMI		kg/m^2^	34	25.11	3.213	52	24.60	3.203	0.470
Start of symptoms		days	34	<3		52	<3		
Viral load, SARS-CoV2		%	34	100		52	100		
Body temperature		°C	34	37.6	0.39	52	37.6	0.44	0.903
Fatigue	100% patients	A.U.	34	1.77	0.47	52	1.79	0.53	0.403
Headache	85% patients	A.U.	31	1.68	0.47	42	1.76	0.48	0.538
Sore throat	42% patients	A.U.	18	1.500	0.514	18	1.500	0.618	>0.999
Cough	38% patients	A.U.	15	1.467	0.516	18	1.222	0.428	0.266
Pain in muscles	38% patients	A.U.	11	1.636	0.674	22	1.773	0.429	0.314
Runny nose	12% patients	A.U.	5	1.800	0.447	6	1.667	0.516	>0.999
Loss of smell	8% patients	A.U.	2	1.000	1.414	5	0.160	0.548	0.619
Loss of taste	0% of patients	A.U.	0	-		0	-		-
Physical activity		A.U.	34	13.88	3.480	52	14.40	3.234	0.410
Physical activity (daily walk)		min	34	8.824	10.94	52	14.13	15.33	0.120
Decreased attention (d2-test)		%E (errors)	34	18.27	29.72	52	21.63	17.51	0.563
URTI		WI score	34	13.21	4.241	52	12.06	4.425	0.235
QOL		WI score	34	32.62	16.00	52	35.02	15.39	0.248
Blood serum IL-6 (normal level <7 pg/mL)		pg/mL	34	19.50 *	76.43	52	11.89 *	20.46	0.738
D-dimer (normal range from 0.1 to 0.5 mg/L)		mg/L	34	1.085 *	2.033	52	5.94 *	38.75	0.596
C-reactive protein (normal level <5 mg/L)		mg/L	34	12.66 *	12.81	52	18.65 *	25.57	0.791
ALT (normal level <35 U/L)		U/L	34	27.69	19.90	52	27.77	22.15	0.988
AST (normal level <32 U/L)		U/L	34	27.82	24.08	52	27.62	22.41	0.956
Total WBC count, (normal range: 3.6–11.0 × 10^9^ cells/L)		10^9^/L	34	5.453	1.247	52	5.496	1.924	0.907
Erythrocytes, RBC (normal range: 3.8–5.8 × 10^12^ cells/L)		10^12^/L	34	4.764	0.479	52	13.60 *	63.69	0.475
Hemoglobin. Hb (normal range 13.5–17.0 g/dl)		g/dl	34	13.00	1.694	52	13.43	1.823	0.272
Hematocrit, HCT (normal range: 40–50, L/L)		L/L	34	41.31	5.159	52	41.44	6.852	0.465
Platelet Count (normal range 150–380 × 10^3^ cells/μL)		10^3^ μL	34	194.6	45.90	52	204.3	49.57	0.336
Neutrophils count (normal range: 1.8–7.5 × 10^9^ cells/L)		10^9^/L	34	6.386	0.996	52	6.716	10.36	0.147
Lymphocyte count (normal range: 1.0–4.0 × 10^9^ cells/L)		10^9^/L	34	2.590	1.020	52	2.366	9.756	0.309
Monocyte count ((normal range: 0.1–1.0 × 10^9^ cells/L)		10^9^/L	34	1.035 *	2.184	52	0.546	3.552	0.094
Eosinophil count ((normal range: 0.1–0.4 × 10^9^ cells/L)		10^9^/L	34	0.1268	0.111	52	0.0902	0.718	0.124
Basophil Count ((normal range: 0.01–0.1 × 10^9^ cells/L)		10^9^/L	34	0.0478	0.0261	52	0.0424	0.262	0.330

*—over the normal range.

**Table 2 pharmaceuticals-15-01013-t002:** The number of compounds (*n*) identified in the *Andrographis* extract (A), *Eleutherococcus* extract (B), and KanJang (C), and their pharmacological effect expressed as a number of deregulated genes (N) in host cells.

Herbal Extracts	Chemical Composition: Number of Compounds Identified in Extracts, *n*	Pharmacological Effect on Gene Expression in Target Cells: Number of Deregulated Genes in Host Cells, N
A—*Andrographis*	39	207
B—*Eleutherococcus*	35	211
C—Kan Jang combination	74	250
D—Andrographolide	1	626

**Table 3 pharmaceuticals-15-01013-t003:** Schedule of examinations and procedures.

	Treatment							Follow-Up
	Day 1 Screening	Day 3	Day 5	Day 7	Day 9	Day 11	Day 14	Day 21
Doctor’s visits	1 Baseline			2			3	4
Eligibility check/Information	*							
Informed consent	*							
Clinical examination	*			*			*	*
Enrollment and allocation to intervention	*							
Treatment (Kan Jang and placebo)	*	*	*	*	*	*	*	
*Biomarker assessments*								
Body temperature (fever)	*	*	*	*	*	*	*	*
COVID-19 PCR test	*			*			*	*
Blood serum cytokine IL-6 (pg/mL)	*	*		*			*	
D-dimer (mg/L)	*			*			*	
C-reactive protein (mg/L)	*			*			*	
Blood cells count analysis	*			*			*	
ALT/AST	*			*				
*Clinician and observer reported outcomes assessments*								
Cognitive performance (tests for attention and memory): d2 test	*			*			*	*
Wisconsin URS Survey Score	*	*		*			*	
Drug intake accountability							*	
Adverse events				*			*	*
*Patient-reported outcomes assessments*								
Mild COVID symptoms: Fatigue Headache Loss of smell Loss of taste Rhinorrhea (nasal discharge) Cough Pain in muscles Sore throat	*	*	*	*	*	*	*	*
Workout, min	*			*			*	*
Physical activity (questionnaire)	*			*			*	*
Paracetamol intake recording	*	*	*	*	*	*	*	
Rescue medication intake recording	*	*	*	*	*	*	*	

*—days of examinations and procedures.

## Data Availability

Data is contained within the article and Appendix A.

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
