# Peer review of "Efficacy of Kan Jang® in Patients with Mild COVID-19: Interim Analysis of a Randomized, Quadruple-Blind, Placebo-Controlled Trial"

_pharmaceuticals, 2022, doi:10.3390/ph15081013_

Round 1

Reviewer 1 Report

 Dear authors,

The manuscript is very interesting and presents potential to  contribute to therapeutical arsenal against COVID-19. Bellow you will find some suggestions

The scientific name of all cited plant species must contain the authority in the first time cited in the text.

Please, check the follow information: The number of patients with negative SARS-Cov-2 virus test was significantly lower in the Kan Jang group compared to placebo after seven days (difference - 12.6±2%, p = 6.31e-008) and 14 days (difference -10.5±2%; p = 4.57e-006) of the treatment, Figure 3a

Please. Include, in M&M the analysis results from supplement 5, concerning the amount of  main compounds/capsules, meaning, give more details in the paragraph: One capsule of Kan Jang® /Nergecov® (size 0, batch No. 40154, Expiry date: 2024-02) contains 260 mg of A. paniculata native extract SHA-10 (drug-native extract ratio of 4.5-8.1: 1, extraction solvent 70% ethanol), including 15 mg of diterpene lactones (andrographolide and 14-deoxy-11,12-didehydroandrograholide), and 18.75 mg of E. senticosus native extract (drug-native extract ratio of 17-30: 1, extraction solvent 70% ethanol, 0.25 mg Eleutherosides B and E). 

Instead inform only the number of detected compounds (Table 3), please, provide the HPLC fingerprint of each extract and  Kanjang, indicating the main  compounds in each one

Reviewer 2 Report

The authors tried to assess the efficacy of Kan Jang on duration and the relief of inflammatory symptoms in adults with mild COVID-19. There are few problems need to be solved by the authors:

1. How did the authors determine mild to moderate COVID-19 symptoms? The standard of the symptoms needs to be listed in the table.

2. How did the authors measure the disease progression rate? The author needs to mention it in detail.

3. Among all the patients involved in the test, whether they have some basic diseases such as respiratory system diseases, or respiratory system diseases since these diseases can influence the efficiency of Kan Jang?

4. What is the main component of Kan Jang? How did it influence the inflammatory symptoms? The mechanism needs to be discussed in depth.

5. Whether Kan Jang had some negative effects after the treatment in COVID-19 patients? The authors need to discuss it in depth.

6. What are the merits and demerits of Kan Jang compared with other COVID-19 drugs?

Reviewer 3 Report

This study investigated Kan Jang (combination of Andrographis paniculata and Eleuthero-coccus senticosus extracts) and its efficacy in COVID-19 patients, especially in relive the symptoms of mild cases. Overall, the study is well-designed and found some efficacy of Kan Jang for mild COVID-19 patients. This manuscript can be accepted. 

Round 2

Reviewer 1 Report

Dear authors,

I consider that this manuscript version brings much more information and improves the results and discussion.

Reviewer 2 Report

In the revised article, the authors modified the manuscript and figures referred to the comments, and answered the questions comprehensively.

This manuscript is a resubmission of an earlier submission. The following is a list of the peer review reports and author responses from that submission.